# Re-Thinking the Role of Government Information Intervention in the COVID-19 Pandemic: An Agent-Based Modeling Analysis

**DOI:** 10.3390/ijerph18010147

**Published:** 2020-12-28

**Authors:** Yao Lu, Zheng Ji, Xiaoqi Zhang, Yanqiao Zheng, Han Liang

**Affiliations:** 1Dong Fureng Economic and Social Development School, Wuhan University, Beijing 100010, China; luyao9007@whu.edu.cn; 2National School of Development, Southeast University, Nanjing 210000, China; xiaoqizh@buffalo.edu (X.Z.); kf1080123@swccd.edu (H.L.); 3School of Finance, Zhejiang University of Finance and Economics, Hangzhou 310018, China; zhengyanqiao@hotmail.com

**Keywords:** information intervention, information disclosing, information blocking, social network, COVID-19

## Abstract

The COVID-19 pandemic imposes new challenges on the capability of governments in intervening with the information dissemination and reducing the risk of infection outbreak. To reveal the complexity behind government intervention decision, we build a bi-layer network diffusion model for the information-disease dynamics that were intervened in and conduct a full space simulation to illustrate the trade-off faced by governments between information disclosing and blocking. The simulation results show that governments prioritize the accuracy of disclosed information over the disclosing speed when there is a high-level medical recognition of the virus and a high public health awareness, while, for the opposite situation, more strict information blocking is preferred. Furthermore, an unaccountable government tends to delay disclosing, a risk-averse government prefers a total blocking, and a low government credibility will discount the effect of information disclosing and aggravate the situation. These findings suggest that information intervention is indispensable for containing the outbreak of infectious disease, but its effectiveness depends on a complicated way on both external social/epidemic factors and the governments’ internal preferences and governance capability, for which more thorough investigations are needed in the future.

## 1. Introduction

The COVID-19 pandemic has attacked the whole world over the past few months. The key features of the Novel Coronavirus, such as long incubation period, high infectiousness, and asymptomatic transmission, were not perceived at the beginning until they were gradually unveiled [1,2,3,4]. The WHO and governments keep disclosing epidemic information, but the disclosure is based on their own endowments, preferences, and perceptions, resulting in misleading information at least in the early stage of COVID-19 outbreak, such as “Masks work? NO” (quoted from Scott Atlas, the White House coronavirus task force member), “This is a flu. This is like a flu” (quoted from Donald Trump, the president of the US), and “There is some immune system variation with Asian people”(quoted form Taro Aso, the Deputy Prime Minister of Japan), etc. This information failed to alert the public but let their guards down instead. Then, the high mortality rate and emergency announcements subsequently incited widespread fear and exacerbated the epidemic situation. Theoretically, a systematic provision of timely and effective information from the government can mitigate the downsides [5]. However, in the real world, speed entails inaccuracy and cognitive uncertainty that keep government away from accomplishing such a tough mission [6]. Thus, in the early stage of an epidemic with strong externalities like COVID-19, the government’s choice between timeliness and effectiveness of intervention strategies raises a theoretical challenge for the management of urgent public health crisis.

The key to successfully contain the spread of an unexpected disease like COVID-19 is to understand the complicated two-way interaction between the dynamics of disease and those of information (and the human behavior response to information) [7]. Information might either amplify or diminish the public’s response to a risk event, depending on the transmission of risk information and public’s reactions at the time it occurs [8]. At the micro-level, one’s behavior depends on the epidemiological status of the disease, the individual’s knowledge about it (information accessed), misinformation, and the individual’s education and income level [9]. Along with the spread of disease in social life (physical level), information spreads in a virtual network, which brings the awareness of crisis for people [10,11,12], leading them to take preventive measures to stay healthy [13,14]. Therefore, the spread of disease facilitates the spread of information, which in turn inhibits the spread of disease [15,16]. However, on the other hand, people usually get illogical, fail to discern falsity, and disregard the truth during information dissemination [17,18]. Misleading information seems to have a natural disposition to resonate with public opinions, which causes spontaneous misrepresentation in transmission [19]. In addition, discussions on epidemic bring panic [20] and aggravate the harm of the epidemic [21], which will be further exaggerated by social media [22]. Meanwhile, increasing uncertainty about the disease makes people feel loss of control and boost people’s anxiety [23], usually accompanied by psychological distress [24]. Therefore, information is critical to fighting against the COVID-19-crisis [25,26], and improper information management strategy may lead to systematic failure [27].

The government, as the main governing body, is the most critical (information) node in the entire network, since it can intervene in information by “blocking” [28,29] and “disclosing” [5,30]. The minimal blocking implies a free and open information environment, which stimulates information to be widely diffused and induces more people to take self-protective measures [31], but “rumors” might also proliferate at the mean time [24]. Even if (under certain premises) some “rumors” are accurate [7,32,33], it might still interrupt the prevention efforts to the epidemic. It is always believed that government should perform as a central node to disclose accurate and up-to-date information to the entire society, so as to keep the public away from untruthful information and prompt the public to make informed decisions about health protection [30,34]. However, in the real world, governments do face time constraints and the trade-off between being accurate and being up-to-date in terms of information disclosing, which is not considered in classical information theory. Because a highly infectious disease caused by unknown viruses with great externality, such as COVID-19, spreads together with information of varying qualities (truthfulness, accuracy, etc.), it is highly probable that the disease has already contaminated the society before low-quality information is purged. In this case, the government has no way to disclose accurate information in time, resulting in the loss of public trust and raising the doubt of the public on the governing capacity of the government, which will accelerate epidemic outbreak [35,36,37]. Therefore, governments need to not only decide when to inject information into the network, but also whether to follow the tenet that governments do not and should not block information spreading at any circumstance [38].

For information blocking, studies have been conducted in theoretical [39,40,41], in empirical [42,43], in case studies [43,44], and other perspectives. These studies argued that, even if governments have the power to control information [45], they should not do that because free spread of information is essential to welfare-maximizing [38]. This argument is based on two underlying assumptions: (1) publishers are completely competitive to reach an equilibrium of disclosing accurate information; (2) there is no time constraint. These two assumptions do not apply for COVID-19 because, in the age of Internet media, people are not incentive compatible to spread accurate information. Moreover, such highly externalized infectious diseases caused by unknown viruses might have already infected a considerable amount of people before low-quality information is purified, so the government should not simply adhere to the tenet of not blocking information when facing an unknown health crisis [38]. As a result, we will discuss the complexity and diversity in information blocking and broaden current information control theory.

By the discussion so far, we notice that the successful containment of epidemic outbreak relies on the successful management on the information dissemination process, which, however, is hard to achieve in the real world. To better understand the failure in containing the COVID-19 pandemic, we here construct an information-behavior bi-layer model by adding a parallel layer of information transmission to the classical **SI** (Susceptible-Infected) model of infectious diseases. We intend to describe the effect of heterogeneous virtual information (at information layer) on heterogeneous nodes’ behaviors (at physical layer) [46]. The government, as the key node in information network, can influence the entire network through information disclosing and blocking. Based on this, we assign values to key variables such as the medical awareness level of the virus and the public’s health awareness level and then conduct computer simulation experiments under different scenarios. We reveal the pattern of government information intervention based on the simulation results. In addition, we only focus on the emergence stage of the infectious disease, during which the recovery from the infected status, such as self-healing and cure of the disease, is omitted [47]; therefore, the base model is the **SI** model rather than the **SIR** (Susceptible-Infected-Recovery) model.

Based on the bi-layered network model, we explore two main themes: how disease spreads affect information spreads and how information affects the efficiency of controlling the epidemic. We introduce the non-dualism of information and the heterogeneity of nodes’ behaviors into the epidemic model and conduct a simulation to reveal the information intervention dilemma faced by the government between information disclosing and blocking. We find that governments face a trade-off between speed and accuracy in information disclosing; and the optimal strategy is contingent on varying conditions in information blocking. The optimal combination of disclosing and blocking is highly sensitive to the government preference and its governance capacity. Governments that are only responsible for the outcome of intervention will focus unilaterally on the accuracy at the expense of speed; a risk-averse government that intends to minimize the maximum infection rate under uncertain scenarios will impose a more restrictive blocking; and the most restrictive blocking strategy might be the best for governments with lower capability and credibility.

In summary, this paper makes several important contributions to the literature. First, existing studies did not pay sufficient attention to the spread and evolution of rumors during a public crisis [48,49,50], which is considered in our study. We expounded the impacts of information dissemination on epidemic evolution in scenarios with different levels of medical awareness of the virus, public health awareness, and government preferences and credibilities, which complements the research in this field. Second, most current studies regard information and disease transmissions as simultaneously happened and jointly induced by the physical movement of an agent [46], while this is not the case during COVID-19 pandemic as the internet obviates the needs for physical contact in information transmissions [51]. Thus, in our paper, we separate the information and disease transmissions and investigate the impact of heterogeneous information on the individual behaviors and disease dynamics. Third, unlike previous research on government information interventions with known risk [5,28,29,30], ours are on government information interventions with unknown risk. The lack of prior knowledge on the Corona-SARS-2 is the most striking feature of the COVID-19 pandemic, which weakens the usefulness of government action and calls for a reassessment of government information intervention under a crisis environment with high uncertainty. To this end, this paper demonstrates a couple of intervention dilemmas faced by government, which complements the existing theories.

## 2. Methods

### 2.1. The Model of Information Disclosing

The information dissemination system resp. behavioral response system is embedded in the information network resp. physical network. Both networks are given as follows.

Information network: the network has (N+1) nodes, first *N* are individual nodes representing *N* individuals denoted as i,i=1,2,⋯N, and one government information node denoted as *j*. The degree of an individual node *i* is denoted as $y_{i}$, which obeys a power-law distribution, that is, $F\left(y_{i}\right)\propto y_{i}^{-v}$, where F(·) is the *CDF* and $y_{i}$ satisfies $\epsilon \leq 1∕y_{i}\leq 1$, where ϵ is a small constant to avoid the degree to blow up. Degree and degree distribution are concepts used in graph theory and network theory. A graph (or network) consists of a number of vertices (nodes) and the edges (links) that connect them. The number of edges (links) connected to each vertex (node) is the degree of the vertex (node). The degree distribution is a general description of the number of degrees of vertices (nodes) in a graph (or network), and, for random graphs, the degree distribution is the probability distribution of the number of degrees of vertices in the graph, which usually assumes a power-law distribution. Throughout the following analysis, we take v=−1 and ϵ=0.01. The government node *j* (representing real-world government) discloses information to every individual node and can only obtain information from $n_{1}$ ($n_{1}\ll N$) (The notation “≪” means that the number n1 must be far less than the number N.) random nodes. The neighborhood of an individual node *i* is the set of all other nodes (including *j*) it connects with, denoted as $O_{i}$.

Physical network: the physical network has *M* nodes, including $n_{2}$ “special” nodes defined as the “gathering spots”, which predisposes these nodes to this epidemic. $M_{t}$ denotes the distribution of locations of all *N* individuals during period *t*, and $M_{0}$ is the initial distribution that can be viewed as the “home” for every individual (node), thus at the beginning of each period *t* the individuals move from $M_{0}$ to $M_{t}$ and return back to $M_{0}$ at the end of period *t*. Home coordinates $M_{0}$ and gathering spots are randomly assigned and different from each other, so we have $N+n_{2}<M$. Suppose there are $n_{3}$ random nodes, each with identical initial information ξ, who disseminate information at the outbreak of disease; $n_{4}$ random nodes are initially affected by the public crisis, representing the “patient zero”.

Without loss of generality, we unitize the information between 0 and 1. The rules for information dissemination in each period are as follows.

Stage i. *Individual nodes send information to neighbors*. Each node that has information at the beginning of each period sends its information to all its neighbors, so all (N+1) nodes might receive information from others. As information is spontaneously [19], rapidly, and extensively [22] misrepresented during transmission, and most people do not send more accurate information than they receive [17,18], we assume that information gets distorted and misrepresented during each transmission. Thus, the actual amount of information received is $\delta x_{i}$ due to information decay, where δ∼U(0,1), and we assume $x_{i}\in \left[0,1\right]$ without loss of generality.

Stage ii. *Individual nodes receive information from neighbors*. Each node might have multiple information sources, and it merges the information from all its neighbors weighted by their degrees (and including itself). Each individual updates its information based on Equation (Equation 1) at each period before the government intervenes:(1)$$x_{i,t+1}=\frac{\sum_{k\in O_{i}}\delta x_{k,t}y_{k}+x_{i,t}y_{i}}{\sum_{k\in O_{i}}y_{k}+y_{i}}.$$

Stage iii. *The government node censors and screens the information*. The government has a threshold $X_{D}$ once it receives information from individuals (otherwise, the government would not act in this stage), the government will screen out all individuals with above-threshold information at the beginning of current period, among whom the government pinpoints the nearest ones and takes the maximum amount of information they carry denoted as $x_{d}$.

Stage iv. *Government node discloses information*. The government is not able to intervene until it censors and screens the information; thus, there is a lag between receiving information and disclosing, which as we can see in Figure 4e, increases with $X_{D}$. After the lag (otherwise, the government would not act in this stage), the government shall disclose $x_{d}$ to all nodes in each period with a weight of λ, where $\lambda \in \left(0,1\right)$. The higher the λ, the more credible the information.

Stage v. *Individual nodes update information again*. The government intervention switches the updating rule to
(2)$$x_{i,t+1}=\lambda x_{d}+\left(1-\lambda \right)\frac{\sum_{k\in O_{i}}\delta x_{k,t}y_{k}+x_{i,t}y_{i}}{\sum_{k\in O_{i}}y_{k}+y_{i}},$$
which is also the final amount of the information after government intervention. In addition, we assume that the amount of information of initial information holders (those who have information in period 0) is constant, i.e., they do not apply for Equations (Equation 1) and (Equation 2).

In short, in the first period, only a few people disseminate information, which will be randomly decayed in each subsequent period, this process simulates the misrepresentation of information.

Twitter data show that there was a significant heterogeneity in the behavioral response to the COVID-19 epidemic [52]. Some people, once informed about the epidemic, wear a mask and practice social distance to not expose themselves to the virus—while others panicked, herded, and behaved irrationally because of bad news, exemplified by flocking to churches for psychological comfort [53], to supermarkets for daily supplies [54], and taking radical actions like repeated hospital visits [55]. Thus, in this paper, we group the population by susceptibleness to irrational behavior caused by information described by an exogenous parameter—individual threshold $X_{I}$ that distinguishes whether an individual is panic-prone or non panic-prone by comparing it with the amount of information the individual has. An above-threshold (under-threshold) information denotes a (non) panic-prone individual. For a panic-prone node, we assume its probability of going to gathering spots instead of maintaining the original trajectory is $1-x_{\cdot ,a}$, where $x_{\cdot ,a}$ is the amount of information it has. For a non panic-prone node, we assume that its probability of not moving is $r_{\cdot ,N}=x_{\cdot ,a}$. Thus, the behavioral routine is as follows (see Figure 1 for a simplified example): a node moves along with its path with a maximum radius $d_{1}$, and the actual distance it moves obeys a uniform distribution in $\left(0,d_{1}\right)$; this node will randomly choose one of the gathering spots if intending to go to one in this period; every individual node follows this routine, then we have an evolving geographical distribution $M_{t}$ of the population moving in period *t*. The uninfected will contact everyone within the maximum infection radius $d_{2}$ and there is a probability μ of being infected for each contact.

Throughout the simulation analysis, we focus on the impact of three key parameters, the initial information (ϵ), individual threshold ($X_{I}$), and disclosing threshold ($X_{D}$), which are the most important quantities to measure the impact of government intervention on the coupled information-disease dynamics. The initial information is the source of all information, which denotes the medical awareness of the virus; the individual threshold is a parameter to distinguish the population by groups set above, the smaller it is, the higher the level of public health awareness. Disclosing the threshold, chosen by the government, measures its relative priority to speed and accuracy in information dissemination. One of the objectives of our experiment is to ascertain the optimal disclosing threshold. Government prioritizes speed more as its threshold is lower, “0“ means that government discloses the information immediately upon receipt; “1” means that government only discloses completely accurate information.

### 2.2. The Experiments of Information Disclosing

The simulation steps will be: (Figure 2 brief overviews these steps):Generate a random information network and a random physical network, the former illustrates the information relationship between people, and the latter records the coordinates of people $M_{0}$ and gathering spots on the map.Assign values to initial information and individual threshold. The initial information is the source of all information, which denotes the medical awareness of the virus; the individual threshold is a parameter to distinguish the population by groups set above; the smaller it is, the higher the level of public health awareness.Assign values to the disclosing threshold. The disclosing threshold, chosen by the government, measures its relative priority to speed and accuracy in information dissemination. One of the objectives of our experiment is to ascertain the optimal disclosing threshold. Government prioritizes speed more as its threshold is lower.Generate random individual nodes with initial information and random initial infected nodes.Enter period 1.
(a)Each individual node with information sends out information to neighbors.(b)Each individual node will update its information (weighted) based on Equation (Equation 1).(c)The government initiates a censoring and screening and enters stage **d** after a lag period, only for the first time does it receive the above-threshold information. If the government never receives above-threshold information, skip **c**, **d**, and go to **e**.(d)Government discloses information to the public, which induces another round of information update for individual nodes based on Equation (Equation 2).(e)The population is grouped into infected and healthy people by health status, and into panic-prone and non panic-prone by how much information one has compared with the individual threshold.(f)Each individual node moves in a physical layer following the routine of the subgroup it is in with probability based on its final information.(g)Reset the infection status of healthy individual within the transmission radius of an infected one according to the infection probability.Return to step 5, initiate a new round for 50 times, that is, run the experiment for 50 periods. The data show a stability after 40 periods, so we stopped at 50.Output the final overall infection rate at the end of period 50.Repeat steps 4–7 for 50 times to reduce the randomness, record the mean, and standard deviation of the final infection rate.Reassign for the disclosing threshold discrete values that equally divide the interval $\left[0,1\right]$ into 11 parts, and repeat steps 3–8 for each value, that is, 11 times, to find the final infection rates for different disclosing threshold scenarios.Reassign for initial information a discrete array 0.4,0.6,0.8,1.0, and reassign for an individual threshold the same values reassigned for the disclosing threshold in the previous step. Then, repeat steps 2–9, that is, 44 times.

Now, we have conducted an experiment with a full parameter space for each initial condition. A total of 484 different conditions were simulated for 24,200 repetitions of the experiment, each lasts for 50 periods, which adds up to a total of 1,210,000 periods of experiments. They essentially cover all possible scenarios under different external constraints. Table 1 lists the definitions, values, and distributions of all parameters in the model.

### 2.3. The Model and Experiments of Information Blocking

We assume the government will suppress any transmission of information under $X_{B}$; thus, $X_{B}=0$ denotes the special case in the previous discussion. Combining the blocking threshold with other initial conditions above, we have a new parameter space simulation with a total of 5324 different scenarios simulated and a total of 266,200 repeated experiments. Each group lasts for 50 periods, for a total of 13,310,000 experiments.

## 3. Results and Discussion

### 3.1. Modeling Framework

Our model consists of two main systems: information dissemination system and behavioral response system. In the information dissemination system, each individual sends (receives) information to (from) its neighbors through an information network. Given that information will always be rapidly, extensively [22], and spontaneously [19] misrepresented during transmission, and that most people do not send more accurate information than they receive [17,18], we assume information gets distorted and misrepresented during each transmission. In the behavioral response system, each individual makes a move according to its information (with probability). Once informed about the epidemic, some people behave rationally such as practicing social distancing, while others behave irrationally such as flocking to churches [53], to supermarkets [54], and taking radical actions like repeated hospital visits [55].

The information dissemination system affects the behavioral response system. The government might intervene in the information dissemination to reduce infections by either disclosing or blocking information.

For information disclosing, the government discloses information to all individuals to make them behave rationally (or at least not behave irrationally). Obviously, the more accurate the information is and the earlier it is disclosed, the public can be better guided which lowers the final infection rate. However, it takes time for government to censor and screen information before disclosing, which brings an accuracy-speed trade-off. We use a disclosing threshold to measure the government’s preference on speed or accuracy; a higher threshold indicates a higher preference on accuracy: a threshold “1” means the government would not disclose any information unless it is completely accurate; while “0” indicates an immediate disclosure without any censoring and screening.

For information blocking, the government blocks less-accurate information transmissions between individuals. Obviously, a stringent blocking leads to a transmission of information with higher accuracy. However, blocking will slow down the overall information dissemination in the network, and then slow down the government’s censoring and screening of information. Thus, there is a trade-off between disclosing and blocking. We use a blocking threshold to measure the blocking stringency, the government would block any transmission of any under-threshold information: a threshold “0” means no blocking at all; while “1” means that government blocks all information transmissions.

We analyze both of the optimal thresholds for government. More details on the settings of our model can be found in the Methods section. In reality, the government has a great influence on the information dissemination. Thus, in our model, we assume that the government node is the most critical one and the government-disclosed information highly outweighs individuals’ information (except for the further discussion of a government with low credibility in a later section).

### 3.2. Intervention Dilemma in Disclosing Information

In this part, we will discuss the speed-accuracy trade-off results and analyze the mechanism in information disclosing.

First, Figure 3 summarizes the results of the simulations with 44 different external constraints, and we find that there is seldom a single dominant disclosing threshold (the government’s preference on speed or accuracy), i.e., seeking either speed or accuracy alone will not result in the lowest infection rate, and the optimal strategy (corresponding to the lowest infection rate at the end of the last period) is somewhere in between, which implies a speed-accuracy trade-off.

Specifically, the optimal disclosing threshold lies between both ends in about 84.09% of the cases, and their distributions vary in different external constraints. Figure 3a,b show that (1) if the initial information is 1 or 0.8 and the individual threshold is in [0,0.7], the optimal disclosing threshold has a 91.75% probability of being in the middle, and mostly (66.89%) falls within [0.6,0.8]; (2) if the initial information is 0.6 and the individual threshold is in [0,0.5], the government has a 93.00% probability of dealing with a trade-off (Figure 3c), and the optimal disclosing threshold mostly (77.78%) falls in [0.4,0.6]; (3) if the initial information is 0.4, the optimal disclosing threshold will almost certainly be greater than 0.4 (99.98%), but the distribution is too scattered to give a specific interval (Figure 3d). The mode of optimal disclosing threshold is 0.8 but only with 20.30% frequency.

In addition, we can see that, if the virus is medically well-known (ξ≥0.8) and public health awareness is low ($X_{I}\geq 0.7$), the government shall prioritize accuracy over speed; if the virus is medically medium-known (ξ=0.6) and public awareness is high ($X_{I}\leq 0.5$), the government shall balance speed and accuracy, which almost equally signifies; and if the virus in medically less-known (ξ=0.4), the government shall probably prioritize accuracy over speed.

Second, we will dissect the underlying logic and mechanism of the government’s trade-off. We take one of the curves in Figure 3a that is denoted by ξ=1 and $X_{I}=0.5$, as an example, to find the relationship between disclosing the threshold infection rate, then we have Figure 4. In all 550 (11×50) experiments, the disclosing threshold for the lowest final infection rate usually lies between 0.7 and 0.9, and the final infection rate first falls then rises as the disclosing threshold increases, with the inflection point being at 0.8 (Figure 4a); the amount of final information per capita and the duration of government intervention both increase monotonically with the disclosing threshold (Figure 4b,e); the number of people infected after government intervention, the number of people infected by a panic after government intervention and the number of uninfected people remaining at the time of government intervention all negatively correlate with the disclosing threshold (Figure 4c,d,f).

One of the fundamental reasons for the government to balance speed and accuracy is the precipitous fall in the marginal contribution of accuracy as the disclosing threshold exceeds a certain “point”, while speed hardly affects the final infection rate. The following is a detailed analysis on the effects of both accuracy and speed.

With respect to accuracy, the effect comes from two perspectives: (1) accurate information lowers the infection from panic in a healthy panic-prone population (Figure 4d); (2) accurate information changes the behavior routine in the panic-prone population, which reduces the spread of the disease. Both (1) and (2) are in play until the disclosing threshold exceeds 0.5 ($X_{I}$ in our example); after that, there is no longer a panic-prone group, neither is infection from panic, which explains the precipitous fall in the marginal contribution of accuracy.

When it comes to speed, the effect comes from two perspectives as well: (1) the time the government spends on censoring and screening information, which we call a lag; (2) the number of remaining uninfected people at the time of government disclosing information. The more accurate information the government seeks, the longer the lag (Figure 4e) and the fewer uninfected people remain at the time of disclosing (Figure 4f). Notice that both have roughly the same slope with respect to disclosing threshold, which explains a roughly constant marginal cost of pursuing accuracy.

The combination of constant marginal costs and abrupt fall in marginal benefits leads to an inflection point in disclosing threshold, which explains the heterogeneity in the distribution of optimal disclosing threshold: as disclosing threshold exceeds individual threshold, a sudden fall of benefits occurs, which theoretically makes the optimal disclosing threshold slightly greater than the individual threshold, which explains what we discussed above that the intervals in which the optimal disclosing threshold mostly lays differ.

In most cases, the government’s premature disclosing of inaccurate information will contaminate the overall network, while obsession with accuracy may have the government miss the disclosure window before too many people are infected, which is intolerable to government who requires a low infection rate. Thus, there is a trade-off.

### 3.3. Intervention Dilemma in Blocking Information

We assume the government will block any transmission of under-the-blocking-threshold ($X_{B}$) information between individuals; thus, $X_{B}=0$ denotes the special case in previous discussion. Other settings are the same as above. In this part, we will discuss the optimal blocking strategies and analyze the mechanism.

As shown in Figure 5, the optimal blocking threshold varies from case to case. Overall, a small blocking threshold ([0.1,0.3]) is necessarily (100%) not optimal; a strict blocking threshold ($X_{B}\geq 0.8$) is usually (50.41%) optimal, experimental data show a value between 45% and 55% in most external conditions; but 0 is the optimal threshold still in 20.25% of cases, and usually (89.80%) occurs when ξ≥0.8 and $X_{I}\leq 0.7$. When the initial information is low (ξ≤0.6), not blocking is seldom (0.83%) optimal.

We have our key findings from the above analysis. First, minor blocking is not an option for government because it is dominated by stricter blocking in a deteriorated or being deteriorated information environment and undermines the efficiency of information dissemination in a benevolent environment. Second, in the age of the Internet, information is extremely interconnected and low-quality information is more easily disseminated, thus stricter information blocking might be an option worth considering in the early stages of an outbreak of an unknown infectious disease. Finally, if the virus is well-known at the medical level, plus the public has a certain level of health awareness, free spread of information might improve the situation; while, otherwise, as in the case of COVID-19, governments should intervene in the spread of information in social networks.

From the simulation results, we can see that, in most cases, the optimal strategy will be either highly stringent blocking or free spread. Blocking low-quality information not only increases the overall information of the whole population, but causes side effects under certain external conditions. Thus, not blocking can be an optimal strategy in some cases. In this section, we provide an in-depth analysis of the data and a mechanistic analysis.

Figure 5b reveals in general the optimal blocking threshold negatively correlates with initial information: as initial information drops from 1 to 0.4, the probability of optimal blocking threshold taking 0 will be 52.07%, 27.27%, 1.65%, and 0%, respectively. Figure 5c shows a positive correlation between optimal blocking threshold and individual threshold.

Governments block information mainly by suppressing less-accurate information, but which, once implemented, will slow down the overall information dissemination in the network anyway. Therefore, blocking can neither be too stringent nor too liberal, an optimal one usually lies in between. However, mild blocking is necessarily not optimal as it fails to purify the information environment.

Furthermore, when the virus is well-known (ξ=1), especially when public health awareness is high ($X_{I}\leq 0.3$), not blocking dominates most of the time (69.70%). While stringent blocking ($X_{B}\geq 0.8$) is necessarily not (0%) an optimal strategy because higher-quality information, which helps to slow the spread of the disease with high public health awareness, is also blocked. Thus, when both the initial information and the level of public health awareness are at a high level, not blocking is optimal; otherwise, information that would not cause panic might do now. In addition, when medical awareness of the virus declines, so does the proportion of valuable information, which necessitates blocking as well.

### 3.4. Optimal Intervention under Different Government Types

In previous sections, our study was based on the neutral government assumption that governments only seek the lowest infection rate. However, in reality, a government is not a personalized organization pursuing social optimum because it is often checked by various inside and outside nodes. In addition, government credibility makes a difference as well. In this section, we will discuss the optimal strategy for non-neutral governments and low-credible governments.

An unaccountable government that evades responsibilities would only care for lower new infections after intervention rather than global infections, which digresses from the objective described above. As shown in Figure 6a, the later the government discloses information, the less that will be newly infected after disclosure. There are two underlying reasons: (1) late disclosed information will indeed be more accurate, which reduces the infection rate; and (2) there are less uninfected people at the time of disclosing. Therefore, a blame-evading government would delay the disclosing to avoid being held accountable.

A conservative government that prefers the least error-prone strategy (minimizing maximum loss) rather than the optimal one (the loss minimization strategy) would block all the information (Figure 6b). Since the optimal strategy would not be accessed until all external conditions are fully judged and scrutinized, which is not feasible for COVID-19, complete blocking would be optimal for such a government to avoid the worst case scenario. Our experiment of 484 different scenarios manifests a complete blocking will never lead to the highest infections.

## 4. Conclusions

In this paper, we introduce the non-dualism (by non-dualism, we mean the information is neither absolutely accurate nor absolutely not but partially accurate) of information and the heterogeneity of nodes’ behaviors into the epidemic model and conduct a simulation to reveal the information intervention dilemma faced by the government and to explore the trade-offs among corresponding strategies. Our experiments highlight that:For information disclosing, governments face a trade-off between speed and accuracy. A better medical understanding of the virus and an inadequate public health awareness make accuracy outweigh speed; otherwise, a quick one is better.For information blocking, the optimal strategy is contingent on varying conditions: no blocking is usually optimal for a well-known virus and a higher public health awareness; otherwise, blocking is preferred.The optimal combination of disclosing and blocking is highly sensitive to the government preference and its governance capability. A government that is only responsible for the outcome of intervention will focus unilaterally on accuracy at the expense of speed; a risk-averse government that intends to minimize the maximum infection rate in uncertain scenarios will impose a more restrictive blocking; and the most restrictive blocking strategy might be best for governments with lower capability and credibility.

These findings reveal the complexity in government decision-making about dissemination of disease information: neither allowing free flow of information nor disclosing it as early as possible is optimal. Under extreme conditions, they are even harmful to the goal of controlling disease outbreak. The interaction between information and infectious disease deepens our knowledge about public health crisis governance, enriches the existing theories in public economics and public management, and provides useful social and policy implications.

In reality, some governments are not as capable and credible as assumed. A lower credibility will discount the effects of disclosing information or even annul it, which makes a total blocking optimal as shown in Figure 6c,d. The bankruptcy of government credibility originates in two ways: (1) the government’s past mediocre performance; (2) the public’s inherent belief in “small government”. Meanwhile, a similar experience in the past also affected government responses and effects, as we can see with the horrible painful memories of SARS inducing vigilance for COVID-19 in East Asia countries, while the U.S. and Europe were indifferent in the early stage of this pandemic.

In the preceding discussion, we relaxed one assumption at a time, whereas the government’s preferences are more complex in reality. In a broader context, the government’s preferences (objective function) are affected by two things: the government’s perception and judgment of the epidemic (decision-making base), and the government’s priorities in different objectives (decision-making objectives); both change over time.

This paper also has some limitations. For instance, our discussion focuses mainly on the theoretical mechanisms behind the joint spreading process of information and epidemic, and the proposed intervention strategies have not yet been analyzed with the real-world data. One reason for the lack of empirical analysis is the complex set-up of the bi-layered network model. The information dissemination network and the physical-layer contact network are not precisely observable in the real world, which makes it challenging for acquiring sufficient data for model fitting. On the other hand, the observed infection and information dissemination process are often already intervened in by the government; therefore, it is hard to separate the net effect of government intervention from the ex-post spreading data. Then, it is technically difficult to quantify the key parameters of intervention. To this end, we believe more sophisticated empirical techniques have to be introduced for the implement data-oriented analysis of our model, such as the network reconstruction and the causal detection techniques, which forms a promising direction for future investigation.

## Figures and Tables

**Figure 1 ijerph-18-00147-f001:**
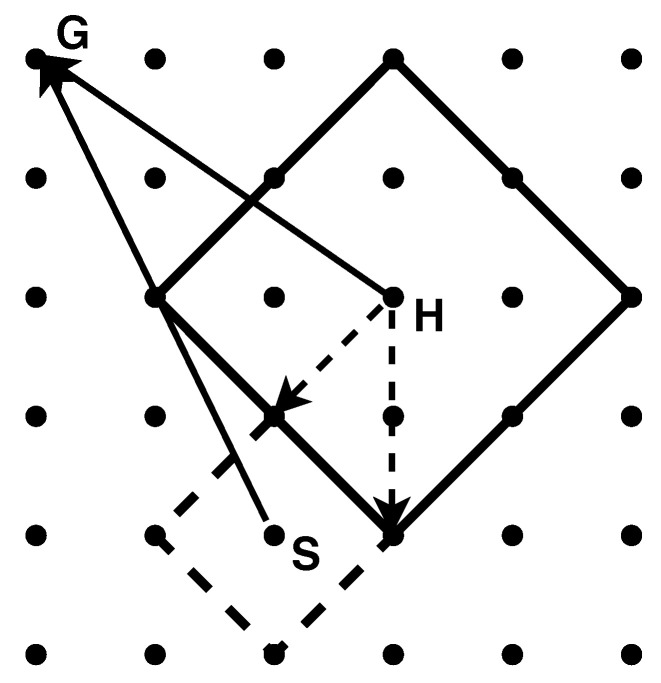
A diagram for individuals’ behavioral routine. Consider a world with 36 nodes ($d_{1}=2$, $d_{2}=1$), among which there are only one infected node (S), one health node (H) and one gathering spot (G).

**Figure 2 ijerph-18-00147-f002:**
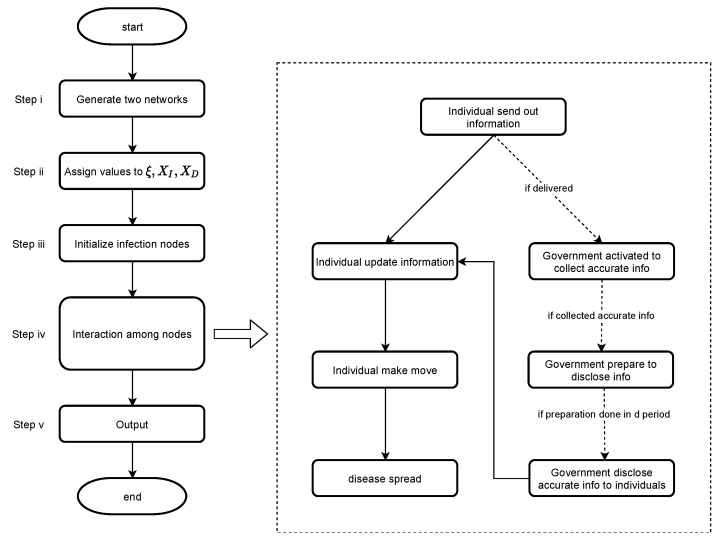
Flowchart of our simulation. In step ii, we assigned 4×11×11=484 different values. Then, we repeated steps iii and iv each for 50 times, respectively.

**Figure 3 ijerph-18-00147-f003:**
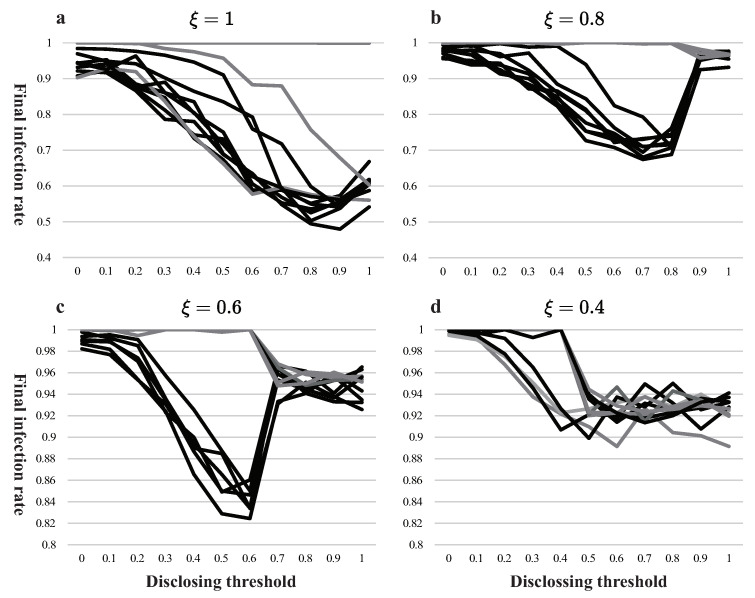
Distribution diagram of disclosing threshold. These four subfigures (**a**–**d**) correspond to initial information of 1, 0.8, 0.6, and 0.4, respectively, each one has 11 curves representing individual thresholds of all 11 values, which shows the optimal disclosing thresholds under all 44 different external constraints. The horizontal axis is all possible values of disclosing thresholds, the vertical axis is the infection rate of the whole society.

**Figure 4 ijerph-18-00147-f004:**
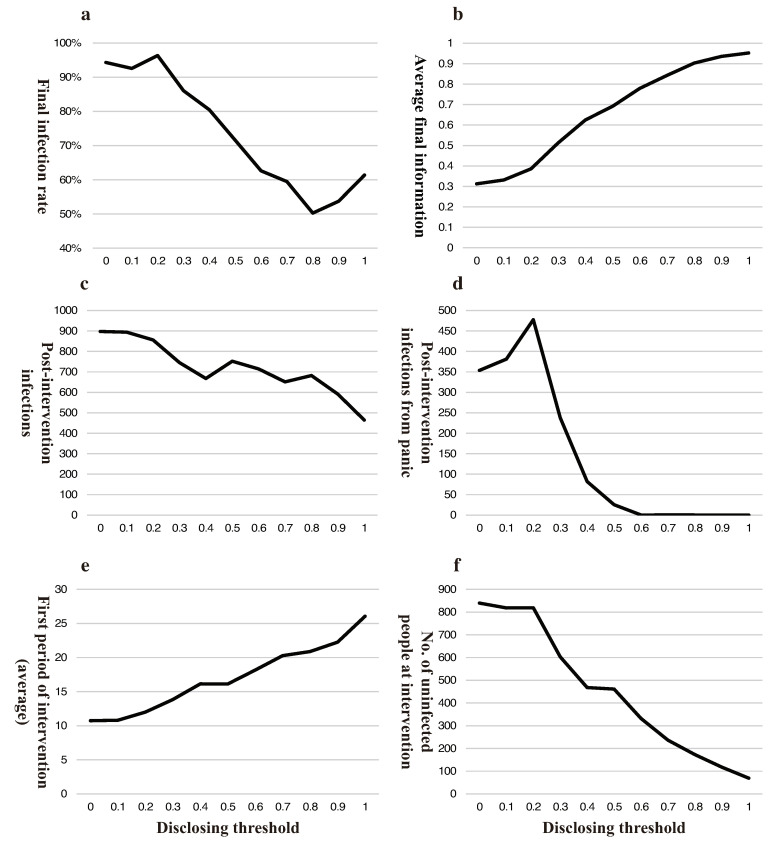
Simulation results for government’s trade-off in disclosing under the special case of $\xi =1,\;X_{I}=0.5$. (**a**) depicts final infection rate; (**b**) conveys the impact of disclosing threshold on information; (**c**,**d**) represent the positive effective of a larger disclosing threshold by showing the new infections overall or from panic after government intervention; and (**e**,**f**) describe the negative effect by showing when the government intervenes and how many health people remained at intervention.

**Figure 5 ijerph-18-00147-f005:**
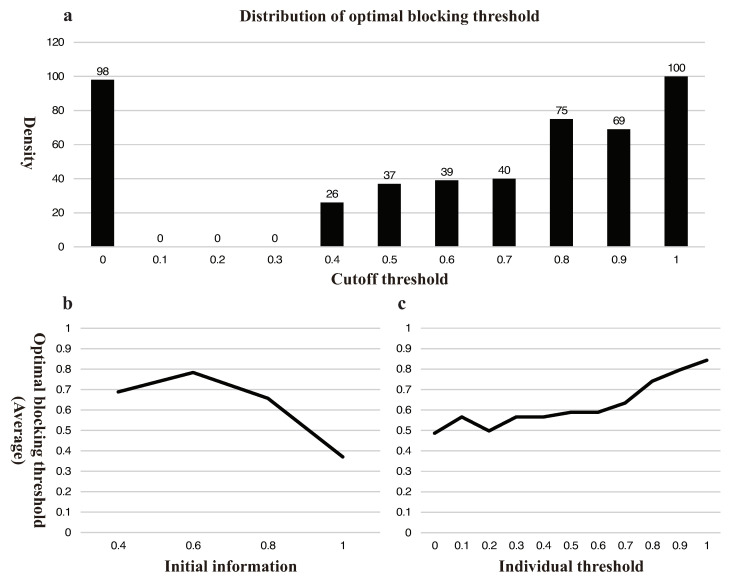
Simulation results for government’s trade-off in blocking. (**a**) conveys the distributions of optimal blocking threshold under 484 different external scenarios (4 pieces of initial information, 11 individual thresholds, and 11 disclosing thresholds). We show how the figure works by taking the first column as an example: there are 98 conditions in which 0 is the lowest infection rate; (**b**) conveys the relationship between initial information and blocking threshold; and (**c**) conveys the relationship between individual threshold and blocking threshold.

**Figure 6 ijerph-18-00147-f006:**
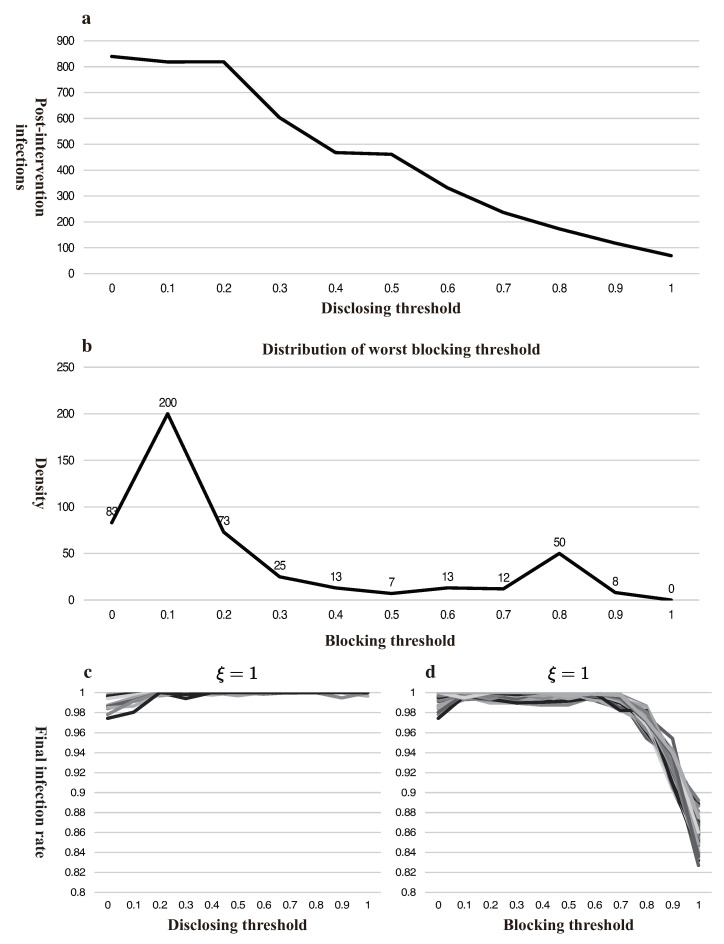
Simulation results for non-neutral governments and low-credible governments. (**a**) portrays the unaccountable governments based on the settings in Figure 4a, the vertical axis is the number of newly infected people after government intervention; (**b**) the distribution of worst blocking threshold (highest infections) based on the settings of Figure 5 portrays the risk-averse government; (**c**,**d**) describe the low-credible governments by reassigning for government’s credibility 10% while keeping other variables unchanged.

**Table 1 ijerph-18-00147-t001:** Definitions, values, and distributions of variables in the model.

Variable	Definition	Values/Distributions
ξ	initial information	0.4,0.6,0.8,1.0
$X_{I}$	individual threshold	0.1,0.2,⋯,1.0
$X_{D}$	disclosing threshold	0.1,0.2,⋯,1.0
$X_{B}$	blocking threshold	0.1,0.2,⋯,1.0
$d_{1}$	maximum moving radius	2
$d_{2}$	maximum infection radius	1
*N*	population	1024
$n_{1}$	population that can send information to government	5
$n_{2}$	numbers of gathering spots	10
$n_{3}$	population with initial information	1
$n_{4}$	initial infections	3
*M*	numbers of nodes (area of the whole map)	2500
μ	infection rate of one-time contact	30%
δ	information decay rate	$U\left(1\%,99\%\right)$
λ	government’s credibility	90%

## Data Availability

The data presented in this study are available on request of the authors.

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
