# Peer review of "Re-Thinking the Role of Government Information Intervention in the COVID-19 Pandemic: An Agent-Based Modeling Analysis"

_ijerph, 2020, doi:10.3390/ijerph18010147_

Round 1

Reviewer 1 Report

Dear Authors,

thank you for the opportunity to read this article concerning this interesting subject. Here are my final comments related to your manuscript ID: ijerph-1006415.

Introduction

The objective of the manuscript seems to be “hidden”. I suggest improving its explication.

Here listed publications that could be useful reference documents to support general background

  • Xiang Gao and Jianxing Yu (2000), Public governance mechanism in the prevention and control of the COVID-19: information, decision-making and execution, Journal of Chinese Governance Vol. 5, pp 178-197.
  • Meijer, Albert, Webster, C. William R., and Contributing Authors. ‘The COVID-19-crisis and the Information Polity: An Overview of Responses and Discussions in Twenty-one Countries from Six Continents’. 1 Jan. 2020 : 243 – 274 (https://content.iospress.com/download/information-polity/ip200006?id=information-polity%2Fip200006)

Metodology

Often difficult to read, I suggest improving the style make it more smooth introducing figurative tools (tables, flowcharts, ecc.) to better explain or summarize methods, assumptions and data.

Line 163: disclosing needs capital “D”

Results

I suggest including lines 371-387 in conclusion section before lines 412.

Recommendation: Minor Revision 

Reviewer 2 Report

In this manuscript, Yao Lu et. al introduced the non-dualism of information and the heterogeneity of nodes’ behaviors into the epidemic model and conduct a simulation to reveal the information intervention dilemma faced by the government and to explore the trade-offs among corresponding strategies. their experiments highlight that we need to pay attention to the spread and evolution of rumors during public crisis. We should separate the information and disease transmissions and figure out the impact of heterogeneous information on the human behaviors and disease dynamics. This article is good overall, with clear logic and reasonable conclusions, except for a few minor errors shown below:

  1. Please check the accuracy in line 106, the “n1«N” should be an error. For the “n1” stands for the population that can send information to government”, the “N” stands for population, the “n1«N” should be n1≤N.
  2. The font of some words is inconsistent with other parts of the article, like the word “one” in line95, the word “first” in line186, and the word “unknown” in line423.
  3. The line 223, there is a needless space in front of the word “Given”.
  4. Please check the lines from the line183 to196, the line space looks strange and need to be adjusted.

Reviewer 3 Report

I would like to congratulate the authors for their work, developing an interesting model for a current and very interesting topic.

Throughout their research, the authors have developed a bi-layer network diffusion model for the intervened information COVID-19 disease dynamics and conduct a full space simulation. Through this model they can show the dilemma faced by organizations and governments between disclosure and blocking of information.

The analysis and development of the document as well as its elaboration are adequate although there are some points that should be reviewed:

  1. Line 23. Authors make a number of statements associated with information transmitted by organizations and governments. It is desirable that authors cite the source of such information so that it can be cross-checked and referenced so that future readers can easily find it.
  2. Authors have not cited the limitations of this model nor its future prospects. It would be desirable to include such limitationes in the final document.
  3. The definition of the theoretical model is adequate but has not been contrasted with reality to verify its reliability. It would be convenient to carry out this verification or, in case it is not possible within this article, to include this aspect within the discussion as one of the considerations and limitations.
  4. Line 382. Authors refer to the actions of various administrations and organizations but do not justify these statements, so that they become personal appraisals lacking scientific rigor. It would have been appropriate to have justified these statements either according to the model presented or according to other publications

Reviewer 4 Report

Dear Authors,

Thank you so much for presenting this interesting topic, just please improve the English level.

Thank you. 

Reviewer 5 Report

This is an important contribution covering an issue of government information management during Corona pandemic from the Chinese perspective.

I believe it is worthy to be published with MDPI.

Evidence base should be substantially expanded.

It is major bottleneck weakness of the manuscript.

Thus I believe diversification of citation track record would much better support claims in the text and add mehtodological reliability to this study.

Thus I would like to warmly recommend consideration for inclusion of few sources beneath alongside others at authors own disposal:

Pisano, G. P., Sadun, R., & Zanini, M. (2020). Lessons from Italy's response to coronavirus.

https://www.mdpi.com/1660-4601/17/24/9404

Christensen, T., & Lægreid, P. (2020). Balancing governance capacity and legitimacy‐how the Norwegian government handled the COVID‐19 crisis as a high performer. Public Administration Review.

https://www.tandfonline.com/doi/full/10.1080/14737167.2020.1823221

Sibley, C. G., Greaves, L. M., Satherley, N., Wilson, M. S., Overall, N. C., Lee, C. H., ... & Houkamau, C. A. (2020). Effects of the COVID-19 pandemic and nationwide lockdown on trust, attitudes toward government, and well-being. American Psychologist.

https://www.ncbi.nlm.nih.gov/pmc/articles/PMC7553250/

Erikson, S. (2020). Pandemics show us what government is for. Nature Human Behaviour, 4(5), 441-442.   Greer, S. L., King, E. J., da Fonseca, E. M., & Peralta-Santos, A. (2020). The comparative politics of COVID-19: The need to understand government responses. Global public health, 15(9), 1413-1416.
